# Experimental and Analytical Study on Creep Characteristics of Box Section Bamboo-Steel Composite Columns under Long-Term Loading

**DOI:** 10.3390/ma14040983

**Published:** 2021-02-19

**Authors:** Shixu Wu, Keting Tong, Jianmin Wang, Yushun Li

**Affiliations:** 1School of Civil and Environmental Engineering, Ningbo University, Ningbo 315211, China; wushixu0928@163.com (S.W.); tongketing@126.com (K.T.); wangjianmin@nbu.edu.cn (J.W.); 2College of Civil Engineering & Architecture, Qingdao Agricultural University, Qingdao 266109, China

**Keywords:** composite column, long-term loading, creep characteristic, creep coefficient, creep model

## Abstract

To expand the application of bamboo as a building material, a new type of box section composite column that combined bamboo and steel was considered in this paper. The creep characteristics of eight bamboo-steel composite columns with different parameters were tested to evaluate the effects of load level, section size and interface type under long-term loading. Then, the deformation development of the composite column under long-term loading was observed and analyzed. In addition, the creep-time relationship curve and the creep coefficient were created. Furthermore, the creep model of the composite column was proposed based on the relationship between the creep of the composite column and the creep of bamboo, and the calculated value of creep was compared with the experimental value. The experimental results showed that the creep development of the composite column was fast at first, and then became stable after about 90 days. The creep characteristics were mainly affected by long-term load level and section size. The creep coefficient was between 0.160 and 0.190. Moreover, the creep model proposed in this paper was applicable to predict the creep development of bamboo-steel composite columns. The calculation results were in good agreement with the experimental results.

## 1. Introduction

Urbanization has developed rapidly in recent years, and the construction industry is developing unprecedentedly. However, some environmental problems produced in the process of construction remain to be solved, such as carbon dioxide emissions, harmful gases pollution, construction waste pollution, and so on [1]. Traditional building materials such as concrete consume a huge amount of energy in the production process, and the traditional building structure is heavy. The destruction and collapse of the bulky structure will cause heavy casualties [2,3]. The construction industry, which is a huge source of carbon emissions, is facing the challenge of sustainable development [4]. Therefore, it has become a trend to develop environmentally-friendly building materials and lightweight structures in the construction industry today [5,6,7,8,9].

Bamboo resources are quite abundant, and bamboo has the advantages of short growth periods and excellent mechanical properties. Meanwhile, bamboo has the advantages of light weight and high strength, and its strength ratio is higher than wood and ordinary steel [10]. It has good conditions as a building material for engineering structures. With the help of modification technology, a variety of bamboo plywood adapted to modern engineering structures can be produced from small-diameter hollow raw bamboo. Bamboo, as a new environmentally-friendly building material, has attracted great attention at home and abroad [11,12,13,14]. However, I-shaped and box-shaped section components made of a single bamboo material cannot reach the strength fully. Therefore, it is of great significance for the development of the green construction industry to study bamboo composite members.

Cold-formed thin-walled steel has the advantages of high strength and homogeneous material. However, it has a large width-thickness ratio, and the slender members cannot reach their ultimate strength because of local buckling or overall instability [15]. Through combining bamboo and cold-formed thin-walled steel in a reasonable way, a new type of composite structure can be formed to achieve the advantages of the two materials fully [16]. Thus, the research group of Ningbo University proposed a new type of bamboo-steel composite structure. The bamboo-steel composite structures consisted of a cold-formed thin-walled steel channel and bamboo plywood with adhesive bonding. This combination method can effectively improve the mechanical properties of the composite members. The lateral stiffness of the bamboo plywood can prevent the cold-formed thin-walled steel channel from premature buckling, and the steel can also improve the stiffness of the entire structure.

Current studies on the mechanical properties of bamboo-steel composite structures, such as composite beams, composite columns, composite floors, and so on, have mainly been focused on short-term force performance research in recent years [17,18,19,20]. The results showed that bamboo-steel composite structures had high bearing capacity and stiffness under short-term loading, which could fulfil the criteria of building structures. However, most loads for the buildings are long-term, and the long-term mechanical performance of the structure is the key factor to ensure the reliability of the structure during the base period of design. Under the long-term constant loading that does not exceed the yield point of the material, the phenomenon that the deformation of the material changes slowly over time is called creep. Wood [21] conducted a creep test of Douglas-fir small specimen for the first time in 1947, and proposed the famous Madison curve. After that, Clouser [22] derived the formula of aging theory based on Wood’s test data. This is the first time that the power series formula of aging theory has been used to study wood creep. In a carefully-controlled creep experiment at constant humidity, Hunt [23] confirmed that the creep of wood followed a straight line on a plot of creep against a logarithm of time, and simplified the long-term creep prediction from short-term data. Zhang et al. [24] studied the creep of Korean pine in 1987, and used the Voigt model to describe its creep characteristics. Then, they obtained the number of viscoelastic models of Korean pine by an experiment and calculated the constants under seven different loads. The research team of Hunan University studied the creep of artificial bamboo panels in 2012. Li [25] showed that artificial bamboo panels were similar to wood panels, and the creep characteristics of artificial bamboo panels were in line with general materials. The calculation results of creep theory were consistent with the test. Yan [26] studied the mechanical properties of bamboo-steel composite beams and composite floors under long-term loading. The experiment showed that the overall working properties of bamboo-steel composite structures were excellent under long-term loading, and the composite structures could fulfil the criteria for structural safety and durability in actual projects. Meanwhile, the experiment proposed a calculation method of creep under long-term loading and the influence coefficient of slip on the composite structures. Under the creep effect and slip effect, the theoretical long-term deformations of composite structures were in good agreement with the experimental values. Therefore, it is of great theoretical significance and engineering application value to conduct experimental research on the creep characteristics of bamboo-steel composite structures under long-term loading.

This article mainly focuses on the creep characteristics of box section bamboo-steel composite columns under long-term loading. Eight composite columns, which consisted of bamboo plywood and cold-formed thin-walled steel, were tested in terms of load level, section size and interface type. In the long-term loading test of more than 200 days, the deformations of composite columns and the regular pattern of creep development were observed and analyzed. The creep prediction model of box section bamboo-steel composite columns was proposed based on the test results, and its reliability was verified.

## 2. Experiment

### 2.1. Materials Properties

The box section bamboo-steel composite columns in this paper were prepared with cold-formed thin-walled steel (Ningbo Haishu Shiqi Zhenren Material Management Department, Ningbo, China), bamboo plywood (Xuancheng Hongyu Bamboo Industry Co., Ltd., Xuancheng, China), and structural adhesive. The mechanical properties of steel materials were tested in the structural engineering laboratory. The thickness of steel was 1.5 mm, and the number of samples was three. The yield strength, tensile strength, elastic modulus, and Poisson ratio of steel sheets were measured according to the reference [27]. The mechanical properties of the steel are listed in Table 1.

The bamboo plywood used in this research was processed by a professional factory. The thickness of the bamboo plywood was 15 mm, and the number of samples was 10. The elastic modulus and static bending strength of bamboo plywood were measured according to “Test methods of evaluating the properties of wood-based panels and surface decorated wood-based panels” [28]. The mechanical properties of bamboo plywood are listed in Table 2.

### 2.2. Test Specimens

The composite column consisted of two cold-formed thin-walled steel channels and four bamboo plywood, which were bonded with structural adhesive. The specific manufacturing processes were as follows. First, the interface between steel and bamboo plywood was polished to remove the stains and galvanized layers of the steel and bamboo plywood. Then, the polished steel and bamboo plywood were wiped with alcohol to ensure that the bonding surfaces were clean. After that, the steel channel and bamboo plywood were combined with adhesive. Finally, the specimens were fixed with fixtures to make sure that the two materials were bonded effectively. A section of the composite column is shown in Figure 1. The consumption of structural adhesive was determined according to the size of specimens, and pressure curing at 25 °C. The finished specimens are shown in Figure 2.

The following three parameters were considered in the long-term loading test. (I) Section size: the dimensions of the specimen cross section were 110 mm × 110 mm and 130 mm × 130 mm, respectively. The thickness of the specimen cross section was 16.5 mm, among which the thickness of the bamboo plywood was 15 mm and the thickness of the steel channel was 1.5 mm. (II) Load level: the load levels were 40% Ny and 70% Ny, respectively. Ny, which was determined by a short-term loading test, was called the yield bearing capacity of the composite column. The Ny of specimens with section size (110 mm × 110 mm) was 188 kN, and the Ny of specimens with section size (130 mm × 130 mm) was 220 kN. (III) Interface type: there were two interface types in this test, including the simple interface that was only bonded with structural adhesive and the composite interface with screws based on the simple interface. Furthermore, the screw spacing of the composite interface was 80 mm. The parameters of specimens are shown in Table 3.

### 2.3. Test Methods

The research group designed and produced a self-balancing long-term loading device (Figure 3) for this experiment. The device mainly consisted of four steel sheets, four screw stems and a set of springs for load retention. Several strain measuring points were arranged along the height direction of the specimen to observe the displacements of the specimen and the changes of strain during long-term loading.

The hydraulic jack was used as the loading system in this study, and the displacement meter was used to measure and record its initial deformation. After loading, the jack was removed and the springs were used to maintain the long-term loading. The long-term loading level of the specimen was controlled according to the deformation value of the displacement meter. The schematic of the loading device of the test is shown in Figure 4.

In this experiment, the device and the specimen were placed indoors and observed for 205 days. During the loading period, the range of room temperature was 8.6–34.5 °C, and the relative humidity changed from 52% to 83%.

## 3. Results

### 3.1. Compression Deformation-Time Curves without Initial Displacement

After seven months of experimental observations, the compression deformation-time curves without initial displacement of eight bamboo-steel composite columns (Figure 5 and Figure 6) were obtained. Figure 5 exhibits the creep-time curves of specimens with the constant cross-section dimensions under different load levels and different interface conditions. The creep-time curves of specimens with the same load conditions under different cross-section dimensions and different interface types are shown in Figure 6.

It can be seen from Figure 5 and Figure 6 that the creep deformation trends of eight bamboo-steel composite columns are basically the same. The experimental results showed that the creep development of the composite columns was fast at first. As the loading time increased, the creep speed decreased, and then the creep development became stable after about 90 days.

### 3.2. Compression Deformation-Time Curves with Initial Displacement

In order to study the proportion of creep in the total deformation, compression deformation-time curves including the initial displacement were drawn (Figure 7 and Figure 8). Figure 7 shows the creep-time curves of the same cross-section, and Figure 8 shows the creep-time curves of the same load.

It can be seen from Figure 7 and Figure 8 that the creep deformation is smaller than the initial deformation of the specimens. At the beginning of the long-term loading experiment, the creep deformation of the specimens developed fast. After about 30 days of the long-term loading, the slope of the creep curves began to decrease significantly. Finally, the deformation velocity tended to be zero and the deformation reached the final creep. It can be found that there are no significant differences between the specimen with a composite interface and the specimen with a simple interface in the final creep deformation.

### 3.3. Creep Coefficient

In order to study the influence of the creep deformation of the bamboo-steel composite column on the overall deformation, the creep coefficient was proposed in this paper. The ratio of creep deformation to initial deformation was defined as the creep coefficient, which is shown as follows:(1)$$\nu =\frac{\Delta L_{\mathrm{creep}}}{\Delta L_{el}}$$
where *ν* is the creep coefficient, Δ*L*_creep_ is the creep deformation, and Δ*L_el_* is the initial deformation.

The creep coefficients of the eight composite columns are shown in Table 4. It can be seen from the table that the creep coefficient of the composite interface specimen is similar to that of the simple interface specimen. This shows that the interface type cannot affect the creep coefficient of the specimen. The creep coefficient of the small cross-section specimen is slightly smaller than that of the large cross-section specimen. This is because increasing the cross-sectional area will reduce the initial deformation, and the proportion of creep deformation will improve. The creep coefficient of the specimen under 40%Ny load level is greater than that of the specimen under 70% Ny load level. The increase in the load level also increases the initial deformation of the specimen, resulting in a weakening of the creep influence on the total deformation.

## 4. Discussion

Figure 5 shows that the creep deformation curves of the specimens with different interface types under the same load levels and the same cross-section dimensions (1-b and 2-b, 1-c and 2-c, 3-b and 4-b, 3-c and 4-c) are basically the same. This means that the influence of the interface types with screws on the creep deformation of the bamboo-steel composite columns is not significant. Compare the creep-time curves of specimens, such as 1-b and 1-c, 2-b and 2-c, 3-b and 3-c, 4-b and 4-c. The initial creep deformation and creep rate of specimen group c are greater than those of group b. Furthermore, the total deformations also increase significantly. It shows that the influence of long-term load level is significant on the creep deformation of the specimen. When the applied load increases, the creep deformation of the composite column also increases.

In Figure 6, comparing 1-b and 3-b, 1-c and 3-c, 2-b and 4-b, 2-c and 4-c, it can be seen that under the same load levels and interface types, the creep deformation of the specimen with a large cross-section is slightly smaller than that of the small cross-section specimen. This shows that increasing the cross-sectional area can reduce the creep deformation. The main reason is that increasing the cross-sectional area will increase the content of steel, while the steel can inhibit the creep development of bamboo. The difference in the steel content of the specimens in this study is not great, thus the difference in the creep deformation of the specimens in Figure 6 is not significant.

It can be seen from Figure 7 and Figure 8 that the creep deformation is smaller than the initial deformation of the specimens. Under 70% Ny load level, although creep deformation is obvious, the proportion in total deformation is still not large. The reasons are as follows: (i) the long-term loading ratio, that is, the load level is small. In this paper, the long-term load applied value is determined based on the yield bearing capacity of the specimen, and the ultimate bearing capacity of the bamboo-steel composite column is 1.5–2.0 times higher than the yield bearing capacity. The long-term load applied value is relatively smaller than the failure load. (ii) The creep of the specimen is affected by the steel. The creep of the bamboo material occurs under long-term loading, while the creep of steel does not occur under normal temperature [29]. Therefore, the steel material of the bamboo-steel composite column can effectively inhibit the creep development of the bamboo material, resulting in a reduction in overall creep.

Comparing the deformation curves of the two interface types in Figure 7, it can be found that the initial deformation of the specimen with a composite interface is smaller than that of the specimen with a simple interface, but there is no significant difference in the final deformation. This shows that the bamboo-steel adhesive interface can reduce the initial deformation by tapping the screw, but its creep deformation will not be affected. Comparing 1-b and 1-c, 2-b and 2-c, 3-b and 3-c, 4-b and 4-c, it is found that the initial deformation and creep deformation of the specimens in group c are larger than those in group b. This shows that the long-term load level has a great influence on the initial deformation and creep deformation of the specimen, and the influence becomes more obvious as the long-term load ratio increases.

Comparing Figure 8a with Figure 8b, it is found that the compression deformation under 70%Ny load level is significantly greater than the deformation under 40% Ny load level. Furthermore, the creep deformation of all specimens is small compared with the initial deformation. Comparing 1-b and 3-b, 1-c and 3-c, 2-b and 4-b, 2-c and 4-c, it shows that the initial deformation and creep deformation are reduced by increasing the size of the cross-section, which is due to the increase in the steel content of the specimen. This shows that the creep deformation of bamboo is restrained by steel, which needs to be considered in theoretical research.

The data in Table 4 show that the creep coefficients of the bamboo-steel composite columns under long-term loading are 0.160–0.190. According to ASTM D6815-09 (2015) [30], the ratio of component creep deformation to initial deformation should not be greater than 1.0. Comparing it with the creep of the wood structure, the creep deformation of the bamboo-steel composite column is smaller. The main reason for this is that the existence of steel greatly inhibits the creep development of bamboo, which indicates that the bamboo-steel composite column has a good long-term mechanical performance.

## 5. Theoretical Analysis

### 5.1. Deformation Relationship between Composite Column and Bamboo

The long-term test results show that the creep characteristics of the bamboo-steel composite column are in line with the characteristics of general viscoelastic materials. The viscoelastic creep theory can be used to analyze the deformation of the composite column under long-term loading. Assume that the creep deformation of bamboo is ∆*L*_b_ at time *t*. The steel and bamboo plywood are bonded as a whole by structural adhesive, and the two materials are stressed and deformed together. It transforms into overall deformation ∆*L* (that is, the data measured by the test) at the same moment. This is equivalent to applying a force *N* ($N=\frac{\Delta L_{b}E_{bt}A_{b}}{L}$) on the bamboo-steel composite column, causing the overall structure to deform.

Assuming that the load acting on bamboo is *N*_1_ and the load acting on steel is *N*_2_, the following equation can be established (refer to Appendix A for the specific process):(2)$$\Delta L(t)=\frac{\Delta L_{b}(t)E_{bt}A_{b}}{E_{s}A_{s}+E_{bt}A_{b}}$$
(3)$$\varepsilon (t)=\frac{\varepsilon_{b}(t)E_{bt}A_{b}}{E_{s}A_{s}+E_{bt}A_{b}}$$
where *E**_bt_* is the elastic modulus of bamboo. At this moment, the creep of the bamboo material has already occurred and the elastic modulus has also changed, so the initial modulus of the bamboo material cannot be used. Therefore, the bamboo deformation modulus at time *t* is adopted. *E**_s_* is the elastic modulus of steel. Since the steel does not produce creep deformation at room temperature, its time effect will not be considered here. *A**_b_* is the cross-sectional area of bamboo, and *A**_s_* is the cross-sectional area of steel.

### 5.2. Creep Model of Bamboo

At present, there are many mature models that can be used to analyze the creep performance of bamboo and wood, such as the Maxwell model, Kelvin model, standard linear solid model, Burgers model, etc. The Kelvin model is made up of elastic basic parts and viscous basic parts in parallel (Figure 9). It can obtain its hysteresis elasticity, and can simulate the creep process of bamboo and wood [31]. Therefore, the Kelvin model is used to predict the creep of bamboo-steel composite columns in this paper.

The constitutive equation of the Kelvin model is as follows:(4)$$\sigma =E\varepsilon_{b}+\eta {\dot{\varepsilon }}_{b}$$

Taking *t* as the independent variable and *ε* as the dependent variable:(5)$$\varepsilon_{b}\left(t\right)=\frac{\sigma_{0}}{E}\left(1-e^{-\frac{E}{\eta }t}\right)$$

The above equation is the stress-strain constitutive model of bamboo, and it is also the relationship between bamboo strain and time. From Equation (5), the relationship between the bamboo elastic modulus and time can be obtained as follows:(6)$$\frac{1}{E_{bt}}=\frac{1}{E}\left(1-e^{-\frac{E}{\eta }t}\right)$$
where *E* and *η* are the elastic coefficient and viscosity coefficient, both of which are undetermined coefficients.

### 5.3. Creep Model and Verification of Composite Column

Substituting Equations (5) and (6) into Equation (3) obtains the relationship between the strain and time of the composite column:(7)$$\varepsilon (t)=\frac{\sigma_{0}A_{b}}{E_{s}A_{s}+\frac{E}{\left(1-e^{-\frac{E}{\eta }t}\right)}A_{b}}$$

This equation, multiplied by the height of the specimen on both sides, is the creep model of the composite column: (8)$$\Delta L(t)=\frac{\sigma_{0}A_{b}L}{E_{s}A_{s}+\frac{E}{\left(1-e^{-\frac{E}{\eta }t}\right)}A_{b}}$$
where *σ*_0_ is the initial stress of bamboo, *A**_b_* and *A**_s_* are the cross-sectional area of bamboo and steel, respectively, *E**_s_* is the elastic modulus of steel, *L* is the height of the specimen, *E* and *η* are unknown coefficients, and ∆*L*(*t*) is the creep deformation at time *t*. The initial parameters of the bamboo-steel composite column under long-term loading are listed in Table 5.

Equation (8) contains two unknown parameters, *E* and *η*, which can be calculated from the creep data measured in the long-term test of the bamboo-steel composite column, and its reliability can be verified. In this paper, the long-term observation data of specimen 2-b are selected to solve the unknown parameters *E* and *η* in Equation (8). After linear regression on the test data of specimen 2-b, the results are as follows: *E* = 29,426.2, *η* = 1,414,721, *E/η* = 0.0208. Substituting the above solution into Equation (8) to sort out the creep calculation formula of the bamboo-steel composite column results in the following:(9)$$\Delta L(t)=\frac{0.0848}{0.5448+\frac{1}{\left(1-e^{-0.0208t}\right)}}$$

Figure 10b shows the comparison between the theoretical value and the experimental value of specimen 2-b. The curve in the figure shows that although there are some differences between the calculated creep value and the experimental value, the overall agreement is good. This shows that the obtained undetermined coefficients are reliable as the elastic coefficient and viscosity coefficient of the bamboo-steel composite column creep model. The creep calculation formula of other specimens can be obtained by using the test parameters in Table 5:

Specimen 1-b:(10)$$\Delta L(t)=\frac{0.0848}{0.5448+\frac{1}{\left(1-e^{-0.0208t}\right)}}$$

Specimen 3-b/4-b:(11)$$\Delta L(t)=\frac{0.0801}{0.5647+\frac{1}{\left(1-e^{-0.0208t}\right)}}$$

Specimen 1-c/2-c:(12)$$\Delta L(t)=\frac{0.1503}{0.5448+\frac{1}{\left(1-e^{-0.0208t}\right)}}$$

Specimen 3-c/4-c:(13)$$\Delta L(t)=\frac{0.1403}{0.5647+\frac{1}{\left(1-e^{-0.0208t}\right)}}$$

The comparison curve between the theoretical value and the experimental value of eight bamboo-steel composite columns is shown in Figure 10. The figure shows that there is an error between the theoretical creep value and the experimental value of the specimen within 50–60 days before the long-term loading test. The curve fits well in the middle and late stages, and the calculated creep value is basically consistent with the overall distribution and trend of the experimental value. The impact of creep on the mechanical properties of components is mainly related to the final creep, that is, the distribution of the late creep.

The theoretical values of the final creep of the composite columns were compared with the experimental values, and the theoretical results were in good agreement with the experimental ones. The relative errors and mean squared errors are listed in Table 6.

The creep model established in this paper can simulate the creep distribution and creep development of the middle and late stages of the bamboo-steel composite column. This shows that the creep development of bamboo-steel composite columns predicted based on this model has great theoretical guiding significance for the evaluation of the mechanical properties of composite columns.

## 6. Conclusions

The distribution characteristics of the creep curve of the bamboo-steel composite column show that whether there are screws at the bamboo-steel interface has basically no effect on the creep deformation of the specimen. Increasing the steel content can reduce the creep deformation of the specimen. Furthermore, increasing the long-term load level will increase the creep deformation of the specimen. The creep effect of bamboo-steel composite columns is most significant by the long-term load level.The creep coefficient of the bamboo-steel composite column under long-term loading is 0.160–0.190, which is less than the requirement of the wood structure code. This shows that the overall creep value of the bamboo-steel composite column is relatively small, and the effect of creep on the applicability of the composite column is not obvious. The box section bamboo-steel composite column has excellent ability to resist long-term deformation.The bamboo-steel composite column creep-time relationship expression is established based on the relationship between composite column creep and bamboo creep, and the creep model of bamboo. That is, the creep model of the bamboo-steel composite column is established, and the unknown coefficients in the model are obtained by linear regression analysis on the creep observation data.The calculated creep value of the bamboo-steel composite column is in good agreement with the experimental value in the middle and late stages. This shows that the creep model established in this paper can simulate the creep distribution of the composite column well in the later stage of loading. Therefore, it is reliable to predict the creep development of bamboo-steel composite columns by using this model, and this model has excellent application value and engineering significance for studying the influence of creep on the mechanical properties of specimens.

## Figures and Tables

**Figure 1 materials-14-00983-f001:**
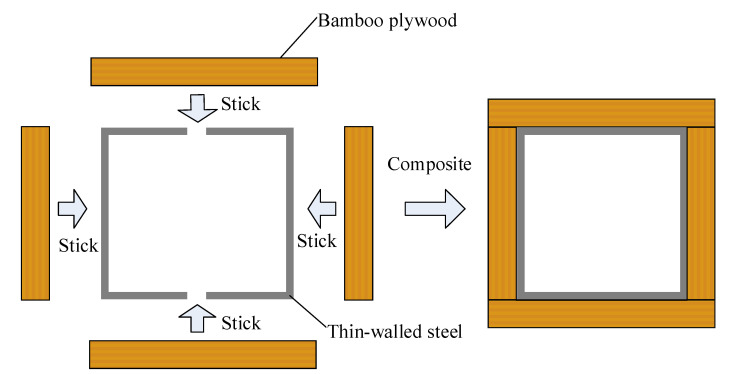
Section of composite column.

**Figure 2 materials-14-00983-f002:**
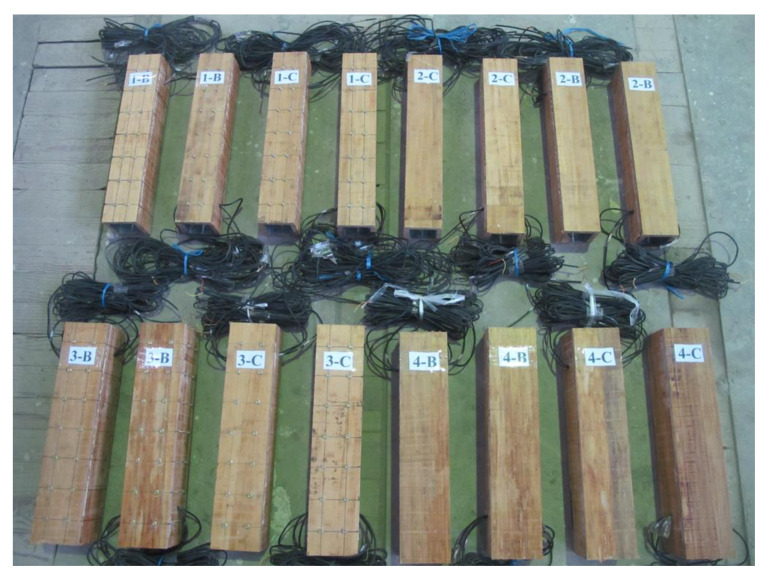
Specimens of composite column.

**Figure 3 materials-14-00983-f003:**
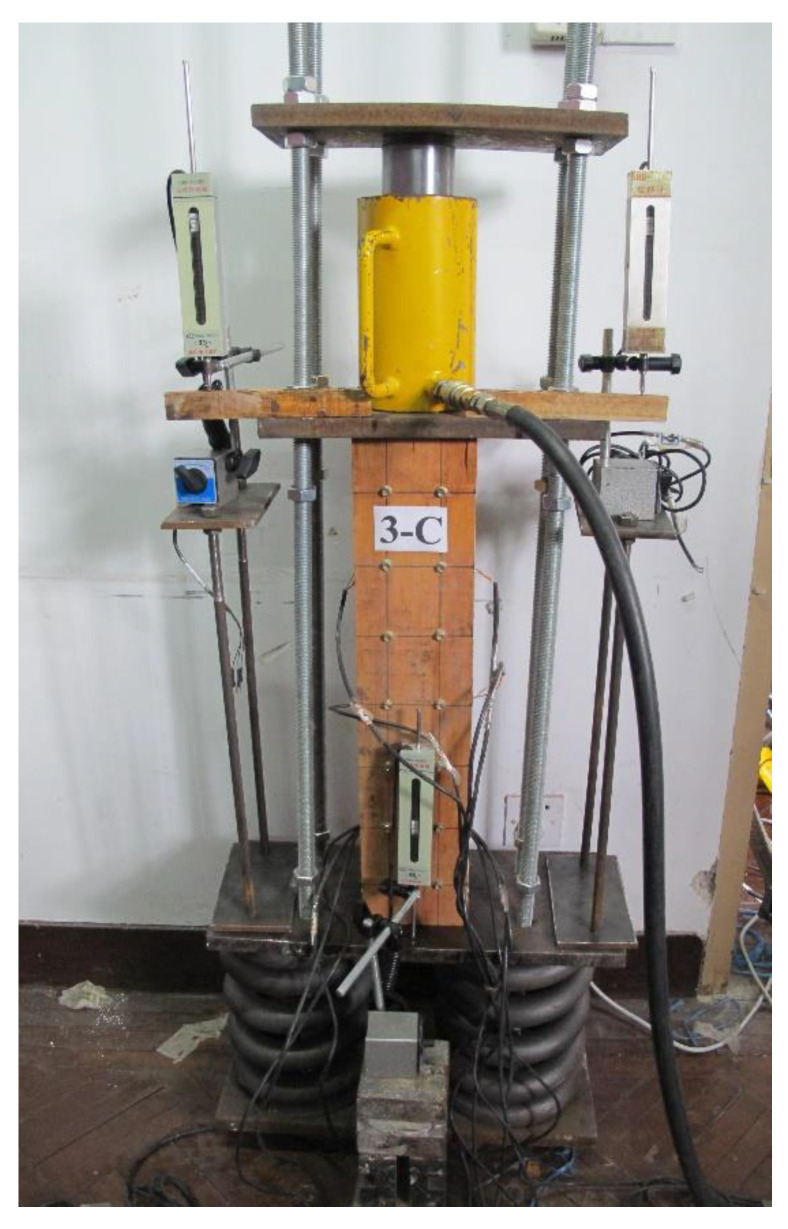
Device for long-term loading.

**Figure 4 materials-14-00983-f004:**
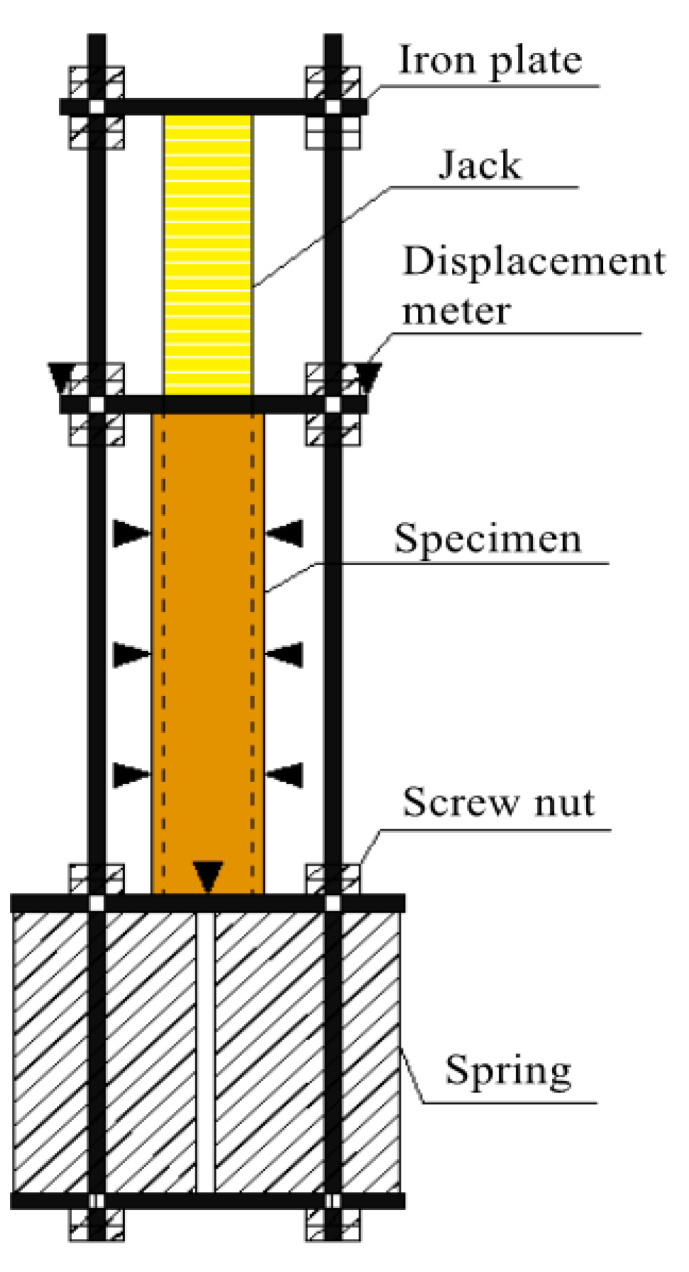
Schematic diagram of loading device.

**Figure 5 materials-14-00983-f005:**
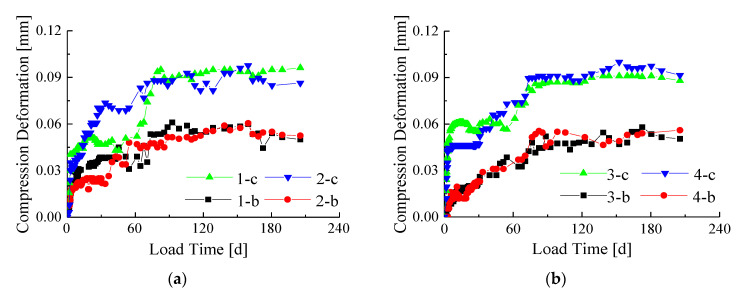
Creep-time curves under the same cross section conditions (without initial displacement): (**a**) Section size of 110 mm × 110 mm; (**b**) Section size of 130 mm × 130 mm.

**Figure 6 materials-14-00983-f006:**
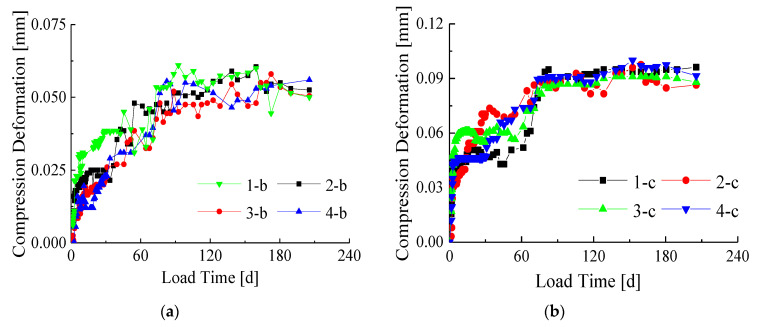
Creep-time curves under the same load conditions (without initial displacement): (**a**) The load level of 40%Ny; (**b**) The load level of 70%Ny.

**Figure 7 materials-14-00983-f007:**
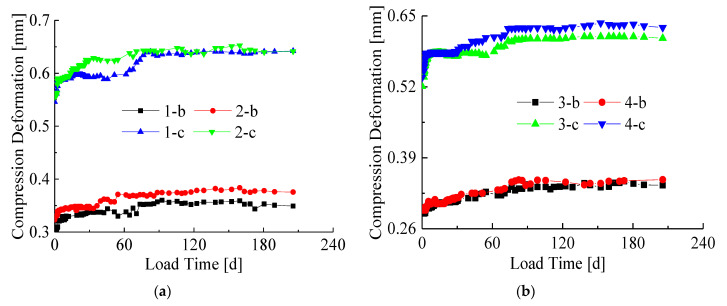
Creep-time curves under the same cross-section condition (with initial displacement): (**a**) Section size of 110 mm × 110 mm; (**b**) Section size of 130 mm × 130 mm.

**Figure 8 materials-14-00983-f008:**
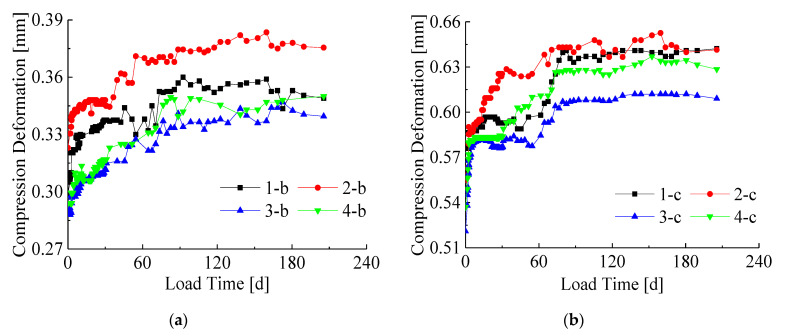
Creep-time curves under the same load level (with initial displacement): (**a**) The load level of 40%Ny; (**b**) The load level of 70% Ny.

**Figure 9 materials-14-00983-f009:**
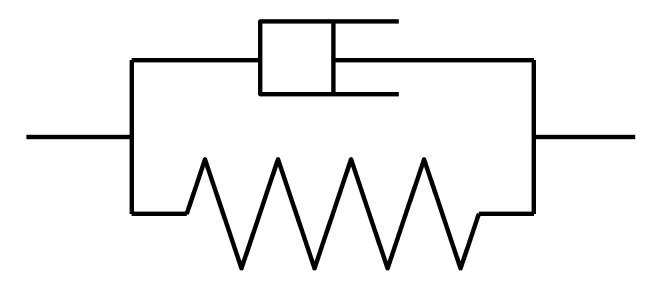
Schematic diagram of Kelvin model.

**Figure 10 materials-14-00983-f010:**
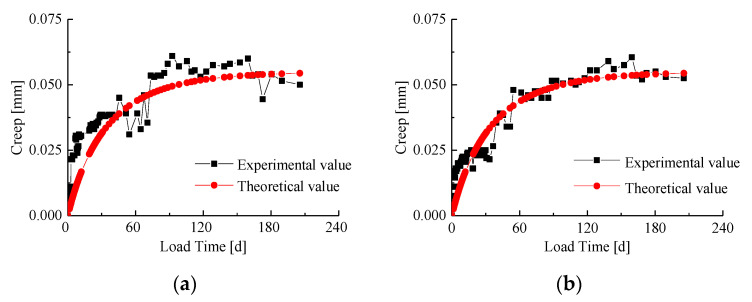
Comparison between theoretical value and experimental value. (**a**) 1-b; (**b**) 2-b; (**c**) 3-b; (**d**) 4-b; (**e**) 1-c; (**f**) 2-c; (**g**) 3-c; (**h**) 4-c.

**Table 1 materials-14-00983-t001:** Mechanical properties of the steel.

Parameter	Yield Strength (MPa)	Tensile Strength (MPa)	Elastic Modulus (MPa)	Poisson Ratio
Average value	261.93	336.68	1.94 × 10^5^	0.271
Standard deviations	1.143	0.858	1.068 × 10^4^	0.014
Variation coefficient	0.44%	0.25%	5.51%	5.17%

**Table 2 materials-14-00983-t002:** Mechanical properties of bamboo plywood.

Parameter	Elastic Modulus (MPa)	Static Bending Strength (MPa)
Average value	7.4 × 10^3^	86.46
Standard deviations	521.447	7.020
Variation coefficient	7.05%	8.12%

**Table 3 materials-14-00983-t003:** Parameters of specimens.

Specimens	Height (mm)	Section Size (mm)	Load Level (kN)	Interface Type	Screw Spacing (mm)
1-b	600	110 × 110 × 16.5	0.4 Ny = 75	Composite	80
1-c	600	110 × 110 × 16.5	0.7 Ny = 132	Composite	80
2-b	600	110 × 110 × 16.5	0.4 Ny = 75	Simple	-
2-c	600	110 × 110 × 16.5	0.7 Ny = 132	Simple	-
3-b	600	130 × 130 × 16.5	0.4 Ny = 88	Composite	80
3-c	600	130 × 130 × 16.5	0.7 Ny = 154	Composite	80
4-b	600	130 × 130 × 16.5	0.4 Ny = 88	Simple	-
4-c	600	130 × 130 × 16.5	0.7 Ny = 154	Simple	-

**Table 4 materials-14-00983-t004:** Creep coefficients.

Specimens	Initial Deformation (mm)	Creep Deformation (mm)	Creep Coefficient
1-b	0.299	0.050	0.167
1-c	0.546	0.096	0.176
2-b	0.323	0.053	0.164
2-c	0.556	0.089	0.160
3-b	0.289	0.051	0.176
3-c	0.521	0.088	0.168
4-b	0.294	0.056	0.190
4-c	0.537	0.092	0.171

**Table 5 materials-14-00983-t005:** Initial parameters of long-term loading test.

Specimens	Bamboo Cross-Sectional Area (mm^2^)	Steel Cross-Sectional Area (mm^2^)	Bamboo Elastic Modulus (MPa)	Steel Elastic Modulus (MPa)	Height [mm]	Load [kN]	Initial Stress (MPa)
1-b	5700	471	7.4 × 10^3^	1.94 × 10^5^	600	75	4.16
2-b	5700	471	7.4 × 10^3^	1.94 × 10^5^	600	75	4.16
3-b	6900	591	7.4 × 10^3^	1.94 × 10^5^	600	88	3.93
4-b	6900	591	7.4 × 10^3^	1.94 × 10^5^	600	88	3.93
1-c	5700	471	7.4 × 10^3^	1.94 × 10^5^	600	132	7.37
2-c	5700	471	7.4 × 10^3^	1.94 × 10^5^	600	132	7.37
3-c	6900	591	7.4 × 10^3^	1.94 × 10^5^	600	154	6.88
4-c	6900	591	7.4 × 10^3^	1.94 × 10^5^	600	154	6.88

**Table 6 materials-14-00983-t006:** Error analysis.

Specimens	Experimental Values [mm]	Theoretical Values [mm]	Relative Errors	Mean Squared Errors
1-b	0.0500	0.0544	8.80%	0.0094%
2-b	0.0525	0.0544	3.62%	0.0037%
3-b	0.0505	0.0507	0.40%	0.0022%
4-b	0.0560	0.0507	9.46%	0.0030%
1-c	0.0962	0.0969	0.73%	0.022%
2-c	0.0885	0.0969	9.49%	0.014%
3-c	0.0880	0.0893	1.48%	0.045%
4-c	0.0915	0.0893	2.40%	0.022%

## Data Availability

The data presented in this study are available on request from the corresponding author. The data are not publicly available due to relating to the publication of the next related paper.

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
