# Peer review of "Experimental and Analytical Study on Creep Characteristics of Box Section Bamboo-Steel Composite Columns under Long-Term Loading"

_materials, 2021, doi:10.3390/ma14040983_

Round 1

Reviewer 1 Report

The paper is very well written. The aim, scope and conclusions were clearly presented. In the reviewer opinion, there is only one point that should be enhanced. Please present the value of load bearing capacity of the columns (Ny). The reviewer is aware of the fact that the behaviour of the steel-bamboo columns under short term loading could be the topic of the next paper. However, some general information are necessary in the present work because the creep of the columns was investigated under the load equal to 40% and 70% of the load bearing capacity (Ny). It is wort to know, whenever the values of the Ny varied regarding the type of the columns. Further, was the failure pattern identical in case of all the types of columns (without or with the screws)? How many specimens were tested under short term loading and what was the CV? Maybe an article already exists that could be cited? Please revised also the parts of the paper after the conclusions, like “Institutional Review Board Statement”.

Author Response

Dear reviewer,

Thank you very much for your comment. I have revised and supplemented the items (highlighted in yellow) in the manuscript. The detailed responses are in the attachment.

Kind regards,
Author

Reviewer 2 Report

The originality and the scientific value of the subject research are good.

The structure of the manuscript is usual, but it must be reworked.
The results, discussion, and conclusion part must be presented separately.

The research area is Experimental and Analytical Study on Creep Characteristics of Box Section Bamboo-Steel Composite Columns.

The current situation of research is insufficiently described. There is a lot of information from the solved research area. It is necessary to state the motivation and benefits of research.

Table 1 and Table 2 - Standard deviations, number of samples, variation coefficient must be added.

Figure 2 must be larger.

Table 3 is on two pages.

Figure 3 is very small. It needs to be supplemented with a description.

Figure 5-8 should be larger.

Formulas 2 - 11 are very elementary - basic. Must be specified in a compressed version.

Overall, the research task is solved logically and clearly.

Figure 10 is on two pages.

It is necessary to evaluate in more detail the resulting approximations - measurement (Figure 10) - mean squared error or another suitable parameter.

Part of the discussion is missing.

It is not possible to publish the manuscript in its current form.

The manuscript must be fundamentally reworked with a better presentation of experiments and research results.

The informative value of the manuscript must be increased overall.

Overall, however, the manuscript can be evaluated positively.

Author Response

(The authors gave the same response as above.)

Round 2

Reviewer 2 Report

Thank you for the adjustments made.
The changes made the improvement of the manuscript.

The research area and results are from the context of the manuscript can better understand.

The results of the research and information value of the manuscript can be evaluated overall well.
However, the presentation of the research result can be further improved.

The manuscript can be published in the journal after adjustments.

Author Response

Dear reviewer,

Thank you very much for your comment. I have revised and improved the presentation of the research results (in red). 

Kind regards,
Author
